# Quantifying the Relationship between Antibiotic Use in Food-Producing Animals and Antibiotic Resistance in Humans

**DOI:** 10.3390/antibiotics11010066

**Published:** 2022-01-06

**Authors:** David Emes, Nichola Naylor, Jeff Waage, Gwenan Knight

**Affiliations:** 1Centre for the Mathematical Modelling of Infectious Diseases (CMMID), Department of Infectious Disease Epidemiology, Faculty of Epidemiology and Population Health, The London School of Hygiene & Tropical Medicine, London WC1E 7HT, UK; david.emes@lshtm.ac.uk; 2AMR Centre, Department of Infectious Disease Epidemiology, Faculty of Epidemiology and Population Health, The London School of Hygiene & Tropical Medicine, London WC1E 7HT, UK; Nichola.Naylor@phe.gov.uk; 3Healthcare Associated Infection and Antimicrobial Resistance Division, UK Health Security Agency, London SE1 8UG, UK; 4London International Development Centre, University of London, London WC1A 2NS, UK; jeff.waage@lshtm.ac.uk; 5Leverhulme Centre for Integrative Research on Agriculture and Health (CGIAR), London WC1E 7HT, UK

**Keywords:** antimicrobial resistance, One Health, agriculture

## Abstract

It is commonly asserted that agricultural production systems must use fewer antibiotics in food-producing animals in order to mitigate the global spread of antimicrobial resistance (AMR). In order to assess the cost-effectiveness of such interventions, especially given the potential trade-off with rural livelihoods, we must quantify more precisely the relationship between food-producing animal antimicrobial use and AMR in humans. Here, we outline and compare methods that can be used to estimate this relationship, calling on key literature in this area. Mechanistic mathematical models have the advantage of being rooted in epidemiological theory, but may struggle to capture relevant non-epidemiological covariates which have an uncertain relationship with human AMR. We advocate greater use of panel regression models which can incorporate these factors in a flexible way, capturing both shape and scale variation. We provide recommendations for future panel regression studies to follow in order to inform cost-effectiveness analyses of AMR containment interventions across the One Health spectrum, which will be key in the age of increasing AMR.

## 1. Background

Antimicrobial resistance (AMR) is an archetypally One Health problem, with increasingly profound global consequences for human health [1,2]. Antimicrobial use (AMU) in food-producing animals considerably outweighs that in humans; and there is evidence of the transfer of resistant bacteria and genes (resistomes) from food-producing animals to humans via the food chain, through direct contact with livestock and indirectly (via contamination of crops and the environment by animal manure and slurry, through waterways in aquaculture, etc.) [3,4,5,6,7,8]. This can contribute to the prevalence of AMR pathogens in humans. Consequently, there have been frequent calls to reduce this usage in order to limit the growth of AMR in humans [5,8,9]. Within this, non-therapeutic AMU in food-producing animals, especially in low- and middle-income countries (LMICs), is seen as a key area for reduction [9,10,11]. However, achieving this reduction can be problematic as antimicrobials are used to increase animal growth (growth-promoting use) and to prevent disease outbreaks among livestock (metaphylactic and prophylactic use), and as such may be important for agricultural livelihoods.

Decision and policy makers now frequently rely on economic analyses when deciding on which intervention to prioritise [12]. In this case, weighing the risks to human health posed by AMU in food-producing animals against its potential economic value is difficult: modellers and policy makers will require a good understanding of the relationship between AMU in food-producing animals and infections due to AMR pathogens in humans in order to justify, prioritise and design interventions to reduce the former [13].

## 2. Reducing AMU in Food-Producing Animals May Not, on Its Own, Be Effective in Reducing AMR Prevalence in Humans

The size and shape of the relationship between food-producing animal AMU and the number of infections with AMR pathogens in humans are uncertain, and there is no consensus on the favourable versus unfavourable consequences of even non-therapeutic animal antimicrobial use [2,14]. For certain bug–drug combinations, genomic studies find that food-producing animal use is unlikely to be an important driver of AMR prevalence in humans. For example, in countries where contact patterns would suggest a greater transfer of AMR (such as living in close proximity to their livestock [15]), studies have contrasting results in terms of directionality and statistical significance [16,17,18,19,20] (see Appendix A for a summary of key studies aiming to quantify this relationship).

Additionally, resistant pathogens pass via the human, animal and environmental components of the One Health system [21], and a high diversity of microbial isolates from across the three One Health compartments suggests that resistance in each compartment has multiple origins [22]. Humans and animals may acquire resistance from exposure to antimicrobials other than those used in animal production, e.g., those used in crop agriculture, or those found in manufacturing effluent and wastewater from production for human use. Especially where resistance is transmitted with ease from humans to food-producing animals, a pool of resistance may be maintained in those animals even when agricultural AMU is low [23,24]. This necessitates a multi-sectoral approach, and suggests that reducing food-producing animal use alone may have only a modest effect on AMR carriage and disease burden in humans [2]. As shown in Figure 1, even in the absence of AMU in livestock, a reservoir of resistant bacteria may be maintained in livestock due to exposure to AMU from the environment and from human use: the One Health system is intertwined.

Some resistant pathogens may already be endemic to human communities independently of transmission from food-producing animals [25,26]. In such cases, resistance prevalence in humans would not respond to the curtailing of AMU in food-producing animals. Using a system of differential equations to model the One Health system, Smith et al. [26]. find that the effect of reducing animal AMU is likely to be much lower when there already exists a high resistance prevalence in humans, for example if resultant from high AMU levels in humans. However, such actions can still limit the emergence of future resistant strains [26].

## 3. Present Quantification of This Relationship Does Not Lend Itself to Cost-Effectiveness Analysis

As agricultural AMU can aid rural livelihoods and poverty alleviation in the short term [27], reducing it implies a trade-off between this and the potential global health benefits of curbing AMR carriage in humans in the medium to long term. This is especially important in highly agrarian economies, and in contexts with a weaker Water, Sanitation and Hygiene (WASH) infrastructure and weaker biosecurity and husbandry practices where antibiotic use may be more relied upon for limiting infectious disease outbreaks in livestock. Although often overlooked, the therapeutic use of antibiotics, which is threatened by AMR, is also important for ensuring animal welfare. Given the theory and literature which suggest that it is still uncertain what effect reducing food-producing animal AMU may have on reducing carriage of AMR in humans (Section 2), the case for such interventions is not clear (aside from a few select drug–pathogen combinations for which a clear link between use and resistance can be established [28]). Aside from the precautionary principle, which argues for the restriction of the use in food-producing animals of antibiotics of critical medical importance to humans, we argue that such decisions, which may have a large impact on farmers, need to be supported by quantification of their likely effect. Without quantifying the empirical relationship between food-producing animal AMU and human AMR, it is impossible to assess the health economic worthwhileness (i.e., cost-effectiveness) of targeting food-producing animal AMU, and thus difficult to make national-level policy decisions about this question.

One way of quantifying the link between food-producing animal AMU and human AMR would be to examine the effect of real-world interventions to reduce food-producing animal antimicrobial use. To this end, the global health community commonly looks to the findings of a meta-analysis by Tang et al. (2017) [29], who synthesised the results of a range of interventions to reduce AMU in food-producing animals. The authors found that, across the thirteen included studies which described AMR outcomes in humans, the pooled prevalence of AMR was 24% lower (CI 6–42%) in intervention groups than in controls, a very large difference. Stratified by the human population considered, the difference was 9% for non-farm workers (CI 5–13%) across three studies, and 29% among farm workers and household members (CI 4–54%) across nine studies.

That meta-analysis is a valuable and novel piece of work that has advanced our understanding of AMR. However, while its results are indicative of the existence of a relationship between food-producing animal AMU and human AMR, they do not allow us to quantify that relationship for the purposes of cost-effectiveness analysis. Included studies looking at human AMR outcomes were pooled across all drug–pathogen combinations and intervention types due to the limited number of studies, which will obscure differing resistance responses of microbial pathogens to AMU [30]. Most studies concerning human AMR outcomes focused on outcomes among farmers and those with direct contact with food-producing animals, with only two included studies looking at the community and two included studies looking at hospital settings (reflective of a major limitation of existing One Health studies in AMR). All included studies looking at human AMR outcomes were from high-income countries, and most were from the Netherlands and Norway, again precluding generalisation.

Moreover, it is difficult to reconcile this large treatment effect with the body of mathematical models and genomic studies which suggest a much more ambiguous relationship between human AMR and food-producing animal AMU (see Appendix A). This leads to concerns of endogeneity, for example that some of the interventions included in the study may have been implemented as part of broader national action plans (NAPs), and thus that the observed effect may also reflect the impact of other contemporaneous interventions to combat AMR (the included studies generally did not control for potential confounders [29]). In a subsequent stratified meta-analysis focusing on food-producing animal AMR outcomes in the same body of literature [31], the authors found that the ostensible effectiveness of interventions was influenced considerably by the underlying prevalence of food-producing animal AMU, again making results difficult to generalise across settings

More broadly, knowing the pooled effect of a set of past interventions presents a black-box relationship between food-producing animal AMU and human AMR. Given that the AMR system functions like a bath with several taps (Figure 1), and given suggestions that reducing food-producing animal AMU alone may be ineffective (Section 2), it becomes necessary to know the effect of altering a range of parameters simultaneously, as well as the nature of their interactions and the role of transmission between compartments. Nor can we know the time scale over which human resistance levels respond to food-producing animal AMU, and any non-linearities inherent in that response, both of which are needed in order to model the whole One Health system effectiveness and cost-effectiveness of food-producing animal AMU interventions.

## 4. Alternative Ways of Assessing This Relationship

Given the difficulty in quantifying the relationship of interest using empirical meta-analyses, we use this section to propose areas of focus for research going forward. We outline appropriate methods for estimating the relationship, and provide guidelines for future analyses. We recommend applying panel regression models to AMR surveillance data, reinforced by transmission dynamic mathematical models of the One Health AMR system (Appendix B).

### 4.1. Transmission Dynamic Mathematical Models

#### 4.1.1. What Is This Method, How Can It Be Used, and How Has This Method Been Used in the Literature?

Transmission dynamic models can take a range of forms, for example including difference equation models or individual-based models (IBMs). Most recent work exploring AMR acquisition in the One Health system has used ordinary differential equation models (ODEs) to estimate the relationship between food-producing animal AMU and human AMR [21,23].

These models track sub-populations over time, with dynamics being a function of the rate of transmission, antibiotic exposure and resistance evolution. For example, Van Bunnik and Woolhouse [23] created a simple set of equations for modelling this system with only two compartments (animal and human), and used Monte Carlo simulation to evaluate the overall sensitivity of human AMR to changes in food-producing animal AMU. Using surveillance data on resistance prevalence and antimicrobial consumption in each compartment, such models can be fitted to specific country contexts to illuminate the relationship between food-producing animal AMU and human AMR in that setting, e.g., Booton et al. [21] for Thailand.

#### 4.1.2. Advantages and Limitations of This Method

Models such as this have the advantage of mechanistically modelling AMR acquisition and transfer using known epidemiological foundations. A main limit to applying such methods is their reliance on detailed, directly-comparable AMR surveillance data [32,33]. For example, the conclusions of the aforementioned paper by Booton et al. are limited by the fitting of the model to Thailand using only ten point prevalence estimates across different One Health settings from hard to reconcile geographically disparate studies [21]. Even where surveillance infrastructures exist, they may be opaque or the data may be siloed [32], and there may be geographical disparities in data availability. In addition, AMR surveillance is rarely set up with a One Health lens in mind, and different design of surveillance systems for humans and animals limits effective comparison. At the time of writing, a number of initiatives are already underway to create AMR surveillance networks [32,33,34,35] and to monitor the costs associated with AMR for the purposes of cost-effectiveness analysis [36].

#### 4.1.3. Future Research Using This Method

Future research should then aim to fit models to longitudinal surveillance data when and where they are available in order to determine the shape and size of the link between food-producing animal AMU (and AMR) and human AMR for policy makers. Because targeting food-producing animal AMU without addressing other factors is unlikely to occur in reality and is potentially of limited efficacy, such models should explore the effect of varying multiple parameters simultaneously (e.g., reducing human–human transmission and food-producing animal AMU in tandem).

### 4.2. Panel Regression Models

#### 4.2.1. What Is This Method, How Can It Be Used, and How Has This Method Been Used in the Literature?

Mechanistic models are limited in their ability to capture and accurately parameterise all the interacting components of a system. Statistical analysis does not rely on knowledge of the mechanisms and instead can determine the contribution of various elements to key dependent variables. For example, panel regression models can be used to combine surveillance data on AMR and AMU prevalence in various countries and/or subnational administrative areas over time, along with other relevant factors such as economic indicators. Here, human AMR prevalence can be regressed against food-producing animal AMU using static methods such as fixed and random effects, difference in difference, or first difference; as well as dynamic methods such as difference and system generalised method of moments (GMM) [37]; and controlling for relevant factors such as medical staffing, portion of employment in agriculture, population density, average annual temperature, and income per capita. Including these diverse covariates allows more meaningful analysis of the various non-epidemiological factors which in mechanistic models may be captured by a single ‘contact’ parameter.

Once panel regression models have been used to determine the link between food-producing animal AMU and human AMR (conditional on covariates), this relationship can then be fed into mechanistic health economic models for the purpose of cost-effectiveness analysis. For this reason, regression models which use data over time (including panel regression) have been recommended over simpler methods such as correlation coefficients for estimating the economic cost of AMR [38].

Panel regression techniques have been used, for example, to assess the relationship between vaccination and AMU in Indian states [39], the relationship between medical staffing and resistance prevalence in Chinese provinces [40,41], the relationship between lab capacity and AMR in European countries [42]; and the association between animal AMR and AMU, and animal AMR and human AMR, in European countries [43].

Such work has been elucidating at the national and subnational level, and works such as Collignon et al. [44] represent a move towards mapping similar relationships at the global level. AMU and AMR data from some countries can be drawn from existing surveillance infrastructures [33,45,46,47,48,49,50]: for other countries, available surveillance data can be used in combination with point prevalence estimates, and missingness partially mitigated using techniques such as multiple imputation [51].

#### 4.2.2. Advantages and Limitations of this Method

A key advantage of panel regression methods is that these models can accommodate flexible functional forms (including lags, interactions, and non-linearities) which can be used to reflect the relationship of interest accurately and comprehensively. They can also control for a diverse range of factors to isolate the effect of each explanator: Noyes, Slizovsky and Singer [52] note that the prevalence of AMR pathogens in humans depends on a number of geographical, economic and social factors which are not generally taken into account in transmission dynamic models but can easily be included in panel regression models as covariates.

By including geographically and economically diverse countries, and by controlling for relevant factors, the fitted values of such an analysis could be used to predict the relationship between human AMR and food-producing animal AMU in various countries based on their characteristics, including those for which no (or few) data are available. In this way, while data missingness is still problematic, it can be overcome relatively effectively in panel models.

Nonetheless, panel models are by no means a magic bullet, nor do the different model types discussed here answer precisely the same question: in fact, panel models can help to quantify the link between AMU and AMR in a way which can subsequently be fed into mathematical models of transmission, as well as cost-effectiveness analyses. The assumptions of panel regression models do not hold when food-producing animal AMU is neither exogenous nor random (as is likely to be the case), and this issue must be addressed carefully (Section 5). In addition to this, underlying causal mechanisms must be understood before applying panel models to data, meaning that panel models must be used in combination with other methods (e.g., impact evaluation and microbiology) to triangulate relationships. As we have noted, cross-country panel regression models with many controls and covariates can overcome data paucity quite well—that being said, this method is still dependent on data availability and will suffer from a lack of comprehensive data.

## 5. Recommendations for Future Analysis

Researchers employing any of the methods discussed here should endeavour to follow (or at the very least consult) appropriate standards and guidelines. For instance, mathematical modelling exercises can make use of the TRACE paradigm [53], can incorporate stochasticity to account for uncertainty and variability, can control for non-linearity and heterogeneity, and can validate the chosen model both internally and externally [54]. The advantage and disadvantages of each method are summarised in Table 1.

As discussed, panel regression studies have the potential to produce much-needed inputs for cost-effectiveness analysis, but only by applying appropriate methods. We therefore recommend that future panel studies aiming to quantify the relationship between AMU in food-producing animals and AMR in humans aim to achieve the following:

Control for social and economic factors as well as epidemiological ones. Collignon et al. [44] make excellent use of social covariates reflecting governance, health system design and the macroeconomy. In particular, covariates which capture contact patterns between humans and food-producing animals (e.g., the portion of employment in agriculture, or population density) should be included. AMU in other animals (such as companion animals) can also be accounted for if data are available.

Use panel rather than cross-sectional regression models. This includes the use of lagged dependent and independent variables to understand the time to effect of the relationship of interest as well as the time dependency of human AMR. Both static and dynamic panel methods should be used and compared [21], allowing researchers to distinguish between the short-run and long-run values of covariates.

Be flexible in terms of functional form. The relationship of interest is a complex one, and not reducible to a single parameter. Because we are unsure of the precise shape of the relationship between human AMR and food-producing animal AMU, we should explore the inclusion of various lags, non-linearities, threshold effects, and interaction terms. Supported by the use of LASSO and other machine learning techniques for covariate selection, this can minimise misspecification and allow the data to ‘speak for themselves’ (although in the case of relative data paucity this may result in the selection of a parsimonious model which ignores epidemiologically important terms [55]). As ‘big data’ on AMR become increasingly available, machine learning prediction models may perform better than linear panel models, especially in identifying and predicting complex causal relationships.

Take statistical potential sources of bias seriously. Regression models can reveal an insignificant or even negative relationship between antimicrobial consumption and AMR [56]. While this may indeed be reflective of reality, such results can also arise from omitted variable bias (e.g., an instance in which wealthier populaces are more likely to be able to afford antimicrobials and simultaneously more likely to have well-funded WASH infrastructures which reduce the spread of resistant pathogens). Such endogeneity can be mitigated by including a range of epidemiologically relevant controls, and by exploring the use of instrumental variable techniques [57] to reflect the exogenous component of AMU. Reverse causality can also become problematic when factors associated with AMR also influence prescribing patterns [58]. Use of one antibiotic can select both for and against resistance to other antibiotic substances within and across antibiotic classes (disjoint and concurrent resistance), and selective pressure also exists from non-antibiotic sources (e.g., heavy metals, disinfectants) in agricultural settings. Thus, using data on the use of one or few antibiotics may not be sufficient. While using data on overall AMU can minimise reverse causality, it may also attenuate any real relationships by including use of antibiotics which do not select for resistance [58]. Given that the use prevalence of various antibiotics may be closely related, multi-collinearity may also become an issue [58].

Include data from LMICs and HICs together. Due in part to data availability, studies on AMR focus disproportionately on high-income settings. The relative social and economic homogeneity of these countries can make it difficult to determine the effect of changes in key parameters when data from LMICs are not included [44]. By understanding the relationship between human AMR and covariates of interest at the global level, the fitted values of such models can be used to predict the relationship of interest even for countries with limited data availability.

Include as outcomes of interest as many drug–pathogen combinations as possible, as well as indices of overall resistance. This will better reflect the unique response of bacterial pathogens to changing levels of AMU [30], in contrast to focusing on only a small handful of drug–pathogen combinations which is commonplace in the literature. Including indices of overall resistance levels will also provide an overview of the total societal effect of food-producing animal AMU, and can avoid the identification problems arising from disjoint and concurrent resistance [59]. An index of overall resistance could, for instance, be the average resistance prevalence from several key drug–pathogen pairs, or an Anderson inverse covariance weighted index [60] of these values which adjusts for correlation between individual outcome measures.

Finally, present results in a form with a cardinal interpretation. For the purposes of comparability, many studies look at simple correlation coefficients or present the effect of a one-standard-deviation change in covariates of interest [44], which do not lend themselves easily to cost-effectiveness modelling. To this end, we recommend that future studies present the effect of a given level of food-producing animal AMU (e.g., in mg per kg of live animal sold, or in mg per human population unit) on the prevalence of colonisation of humans by resistant bacteria. This will not be a single value, but a potentially complex relationship.

## 6. Final Remarks

The two methods discussed here are not mutually exclusive, nor exhaustive, and we recommend that the results of both be compared, along with other methods, in order to triangulate the relationship between food-producing animal AMU and human AMR prevalence. These methods could include randomised control trials, quasi-experimental settings and natural experiments, as well as other modelling strategies such as hierarchical risk-assessment models of AMR acquisition [61].

Parameterising and validating models employing the methods discussed in this article will benefit from the presence of stronger surveillance data on AMR and consumption, across all three One Health compartments. For this reason, we underline the importance of continuing to establish and expand AMR surveillance networks globally.

Just as it is important to quantify accurately the effect of food-producing animal AMU on human AMR, so it is equally important to continue to quantify accurately the effect of food-producing animal AMU on farm productivity and profitability and on animal welfare, as this outcome represents the other ‘half’ of the trade-off implied by reduced food-producing animal AMU. Randomised trials at the farm level should be the gold standard for estimating this. Outcomes considered should include feed conversion rate, animal morbidity and mortality, the cost (labour and medicine) of treating sick animals, and the effect of replacing antibiotics with alternatives such as probiotics [62]. Randomised trials should consult guidelines such as CONSORT [63]. These results can be combined with the effect on human AMR to generate holistic cost-effectiveness analyses of hypothetical interventions to reduce food-producing animal AMU, consulting standards such as the Harvard–Gates guidelines for cost–benefit analysis [12]. This cost-effectiveness analysis will provide a robust basis for policy decisions concerning food-producing animal AMU.

In fact, a complete understanding of the relationship between animal AMU and human AMR will require the syncrasy of AMU and AMR surveillance data, in vitro experiments on antibiotic susceptibility in human and animal isolates, and animal experiments which evaluate the AMR (and other) outcomes of different levels of AMU [64].

AMR is a significant global problem, one which is set to become more important in the near future, and one which may disproportionately affect the Global South [65]. At present, it is understood that AMU by food-producing animals is related to AMR carriage in humans. However, the shape and size of this relationship, and precisely how food-producing animal AMU interacts with other factors such as current AMR prevalence or macroeconomic characteristics, remain largely unknown, and our current understanding of this relationship does not allow us to make that important decision with confidence. In order to inform policy, and in particular to model the effectiveness and cost-effectiveness of interventions targeting food-producing animal AMU as a way to improve human health, a more exact understanding must be reached using a data-driven approach. We suggest ways forward in this article, and outline what we feel is an important knowledge gap.

## Figures and Tables

**Figure 1 antibiotics-11-00066-f001:**
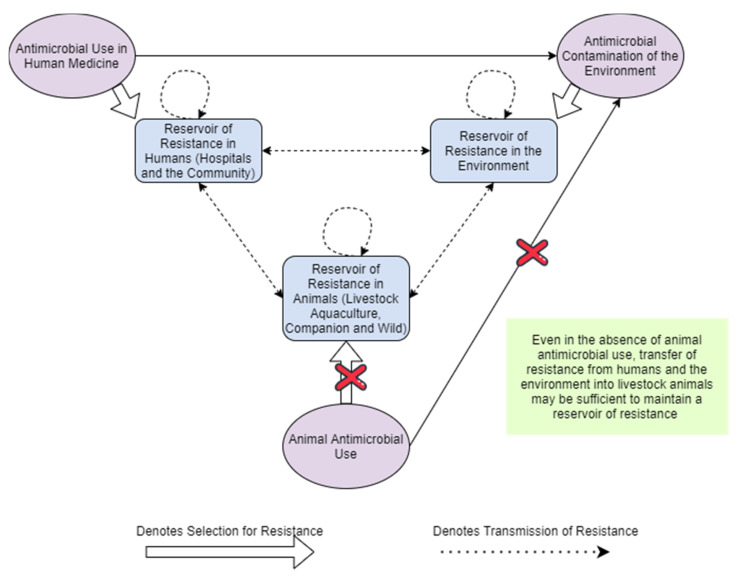
Simplified System Reflecting the Maintenance of Resistance Reservoirs in the Absence of Food-Producing Animal Antimicrobial Use. (Rectangles represent reservoirs of resistance, ovals represent introduction of antimicrobials into the system, and crosses represent the interruption of transmission or selection mechanisms.)

**Table 1 antibiotics-11-00066-t001:** Advantages, disadvantages, and data requirements for different methods of estimating the relationship between animal AMU and human AMR.

Method	Advantages	Disadvantages	Data Sources
Transmission Dynamic Mathematical Models	Mechanistic capturing of AMR evolution Once fitted, can be used to predict/explore scenarios	Requires comprehensive data Complexity of development	Prevalence of AMR in infections in both humans and livestock Multiple country data on antibiotic use across the One Health spectrum
Panel Regression Methods	Accommodation of flexible functional forms Complexity in many factors can be included Overcome data gaps	Has difficulty accounting for exogenous or random relationships Requires causal mechanism understanding Requires comprehensive data	Prevalence of AMR in infections in both humans and livestock Multiple country data on antibiotic use across the One Health spectrum data on social and economic factors

## Data Availability

Not applicable.

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
