# Peer review of "Quantifying the Relationship between Antibiotic Use in Food-Producing Animals and Antibiotic Resistance in Humans"

_antibiotics, 2022, doi:10.3390/antibiotics11010066_

Round 1

Reviewer 1 Report

I would like to congratulate the authors of the article on the relationship between Food-Producing Animals and Antibiotic Resistance in Humans health and quantification. The author has presented a good research work with a good presentation of the results; however, the authors need to concise the article as common facts are discussed repeatedly. 

I would recommend the article could be published in Antibiotics, after a minor revision.

The authors need to address the below-mentioned queries.

1. The author needs to discuss Figure 1 concisely.

2. For the “Food Producing Animals and Antibiotic Resistance in Humans” section: The author could concise the repeated discussion and make it more concise with figures or graphs (e.g. meta-analysis by Tang) to provide a clear picture to the readers and motivation.

3. The authors need to provide at least one set of data for each model discussed in the article.

4. A tabular presentation of the advantage and disadvantages of different methods discussed in this article will be benefited the readers of the article. Appendix A should be clearer, as it does not provide a clear picture.

5. All the references should be in the same format. Some references missing page numbers. The author should follow the journal guidelines while showing references in the text.

6. The author could provide the following relevant references:

(i)Shryock TR. Relationship between usage of antibiotics in food-producing animals and the appearance of antibiotic resistant bacteria. Int J Antimicrob Agents. 1999 Aug;12(4):275-8. doi: 10.1016/s0924-8579(99)00089-8. PMID: 10493602.

(ii) Marshall, Bonnie M, and Stuart B Levy. “Food animals and antimicrobials: impacts on human health.” Clinical microbiology reviews vol. 24,4 (2011): 718-33. doi:10.1128/CMR.00002-11

Author Response

We thank the reviewer for their comments, which we have incorporated into the revised manuscript. Point-by-point responses to comments are detailed in the attached Word document. 

Reviewer 2 Report

This manuscript is not an original article but a review, so I cannot properly make an assessment using this form.

I would like to suggest that the manuscript should be resubmitted according the rules of a review.

Reviewer 3 Report

AMR is a significant global issue that is expected to grow in importance in the near future and that may disproportionately affect the Global South. AMU by food-producing animals is now thought to be linked to AMR carriage in humans.

The authors of this manuscript detailed and compared methods for estimating this relationship, drawing on major literature in the field. The advantages and drawbacks of each method, as well as future study employing each method, are discussed in detail in the manuscript.

The authors attempt to demonstrate that, while mechanistic mathematical models have the benefit of being based in epidemiological theory, they may struggle to incorporate crucial non-epidemiological factors with an uncertain relationship to human AMR. They also advise for further use of panel regression models, which can incorporate these characteristics more easily and capture both shape and scale variation.

They propose recommendations for future panel regression research to guide cost-effectiveness studies of AMR containment methods across the One Health spectrum, which will be crucial in a period of growing AMR.

I would like to recommend the manuscript for publication in the Antibiotics if the authors could revise the manuscript and modify the citation of the references in the text throughout the manuscript. Also, Appendix A and Appendix B are cited in the text as Appendix 1 and Appendix 2.

Author Response

We thank the reviewer for their comments. We have changed the references to Appendix A and B, and edited our citations.
